# Pt Nanoparticles with High Oxidase-Like Activity and Reusability for Detection of Ascorbic Acid

**DOI:** 10.3390/nano10061015

**Published:** 2020-05-26

**Authors:** Qin Cheng, Yong Yang, Yusi Peng, Meng Liu

**Affiliations:** 1Shanghai Institute of Ceramics, Chinese Academy of Sciences, Shanghai 200050, China; cq18817206957@163.com (Q.C.); pengyusi@student.sic.ac.cn (Y.P.); liumeng1@shanghaitech.edu.cn (M.L.); 2Institute of Nanochemistry and Nanobiology, School of Environmental and Chemical Engineering, Shanghai University, Shanghai 200444, China

**Keywords:** Pt nanoparticles, oxidase-like activity, reusability, colorimetric method, ascorbic acid

## Abstract

Noble metal nanoenzymes such as Pt, Au, Pd, etc. exhibit magnificent activity. However, due to the scarce reserves and expensive prices of precious metals, it is essential to investigate their enzyme-like activity and explore the possibility of their reuse. In this work, the oxidase-like activity and reusability of several Pt nanoparticles with different morphologies were detected. We compared the Pt nanoparticles (NPs) with a size of about 30 nm self-assembled by 5 nm Pt nanoparticles and Pt nanoparticles (Pt-0 HCl) with a diameter of about 5 nm, and found that their Michaelis−Menten constants (Km) were close and their initial performance similar, but the Pt NPs had better reusability. This was probably attributed to the stacked structure of Pt NPs, which was conducive to the substance transport and sufficient contact. At the same time, it was found that the size, dispersion, and organic substances adsorbed on the surface of Pt nanoparticles would have a significant impact on their reusability. A colorimetric detection method was designed using the oxidase-like activity of Pt NPs to detect ascorbic acid in triplicate. The limits of detection were 131 ± 15, 144 ± 14, and 152 ± 9 nM, with little difference. This research not only showed that the morphology of the catalyst could be changed and its catalytic performance could be controlled by a simple liquid phase synthesis method, but also that it had great significance for the reuse of Pt nanoenzymes in the field of bioanalysis.

## 1. Introduction

Ascorbic acid (AA), as an antioxidant, plays an important role in many biochemical and physiological processes, such as the formation of neurotransmitters, ion absorption, amino acid metabolism, and so on [1,2]. It also can prevent and treat scurvy, colds, cardiovascular diseases, allergies, and cancer [3,4,5,6]. However, the human body cannot synthesize ascorbic acid through its own metabolism and must ingest it from the outside [1,7,8]. In addition, ascorbic acid is widely present in vegetables, fruits, and medicines [9,10]. Therefore, developing a simple, efficient, and sensitive detection method is very important for these fields. Thus far, mainstream methods for detecting AA have been reported including the electrochemical method, liquid chromatography, fluorescence method, titration, and chemiluminescence technology [11,12,13]. However, these methods all have the disadvantages of complicated and expensive operations. Due to having the advantages of visibility, simple operation, and low cost, the technology of colorimetric detection of small biomolecules, such as ascorbic acid, has become more and more mature and has attracted wide attention [14,15,16,17]. 

In recent years, with the rapid development of nanotechnology, lots of nanomaterials have been found with enzyme-like activities (such as Pt, Au, Fe_3_O_4_, CeO_2_, Co_3_O_4_, etc.) [18,19,20,21,22,23]. Compared to natural enzymes, nanoenzymes have the advantages of simple synthesis, low cost, good stability, etc. [22,24]. At present, the synthesis of nanoenzymes with high catalytic activity and investigations into their mechanisms of action are topical. 

Pt nanomaterials (Pt NPs) have a variety of catalytic properties and are widely used in energy, medicine, chemical, and environmental fields [22,25]. As a kind of nanoenzyme, Pt nanoparticles have the advantages of high activity, great biocompatibility, and broad application prospects [26]. In addition, Pt NPs have been reported to have various enzyme-like activities like oxidase, peroxidase, SOD, polyphenol oxidase, and catalase [22,23]. However, Pt is an expensive precious metal [25,27]. Therefore, it is essential that the Pt NPs are recycled after use in order to conserve supplies. At present, the research of Pt-related nano-enzymes have focused on improving the catalytic activity, while few attempts have been taken to study its reusability. In this work, we found that Pt NPs, as synthesized, were reusable and could catalyze 3, 3, 5, 5′-tetramethylbenzidine (TMB) oxidation multiple times in a row. Based on this, we explored a series of Pt NPs and found that their reusable properties and affinities to the substrate were strongly related to their morphology and surface state. In addition, the method for colorimetric detection of AA was established to detect AA in triplicate, and the limits of detection were 131 ± 15, 144 ± 14, and 152 ± 9 nM, which had broad application prospects. These investigations not only extend our understanding of the properties of Pt NPs, but also provide a significant basis for the development and design of nanoenzymes.

## 2. Experimental

### 2.1. Chemicals

Chloroplatinic acid hexahydrate (H_2_PtCl_6_·6H_2_O) was purchased from Sigma-Aldrich. In addition, polyvinylpyrrolidone (PVP), 3, 3, 5, 5′-tetramethylbenzidine (TMB), and ethylene glycol (C_2_H_6_O_2_) were obtained from Alfa Aesar. Ascorbic acid, acetic acid-sodium acetate buffer solution (0.2 M, pH 4.0), and n-hexane were purchased from Aladdin, China. The water used was deionized water (18 MΩ cm). 

### 2.2. Characterization

The absorbance spectra of oxidase reactions were measured by ultraviolet-visible spectrophotometry (Lamda950), and the morphology of Pt nanoparticles (Pt NPs) was tested by transmission electron microscopy (TEM, Tecnai G2 F20).

### 2.3. Synthesis of Pt Nanoparticles 

In a 50 mL three-necked flask, 2.5 mL of ethylene glycol (EG) was refluxed at 180 °C for 5 min with an oil bath, after which 0.34 mL of HCl (37 wt%) was added. Then, 0.94 mL of H_2_PtCl_6_·6H_2_O (0.0625 M) solution and 4 mL of PVP (0.0375 M) were added into the above solution for 10 times within 5 min. During the whole reaction, the reactants were continuously stirred and mixed uniformly. After 20 min, the three-necked flask was taken out, naturally cooled to room temperature. The sample was collected and washed by centrifugation with water and ethanol. Finally, the samples were dried in an oven at 60 °C. By changing the concentration of HCl (37 wt%) into 0 wt%, 3.7 wt%, and 9.25 wt%, we prepared Pt-0 HCl (0 wt%), Pt-3.7 HCl (3.7 wt%), and Pt-9.25 HCl (9.25 wt%), respectively. In addition, changing the concentration of PVP solution (0.00375 M) into 0.075 M, we prepared Pt-2PVP (0.075 M).

### 2.4. Kinetic Assay of Oxidase-Like Pt Nanoparticles 

First, 60 μL Pt nanoparticles (0.75 mg mL^−1^, dispersant was deionized water) and 140 μL of TMB were added into 2.8 mL of acetic acid-acetate buffer solution (0.2 M, pH 4.0) at 35 °C. The kinetic measurements were performed in a time course mode, and a spectrophotometer was used to measure the absorbance spectra in the first 3 min of the reaction system at 652 nm every 15 s. Lineweaver–Burk graphs were used to calculate Michaelis–Menten constants: *1/v = Km/Vmax(1/{S} + 1/Km).*

### 2.5. The Oxidase-Like Activity of Pt Nanoparticles

First, 40 uL TMB (5 mM) and 60 uL Pt nanoparticles (0.75 mg mL^−1^, dispersant was deionized water) were added into acetic acid-sodium acetate buffer solution (0.2 M, pH 4.0, 2.9 mL), and the solution was incubated at 35 °C with water bath for 10 min. UV-visible absorption spectra were used to monitor the reaction at 652 nm.

### 2.6. The Reusability of Pt Nanoparticles

First, 40 uL TMB (5 mM) and 60 uL Pt NP solution (0.75 mg mL^−1^, dispersant was deionized water) were added into 2.8 mL acetic acid-sodium acetate buffer solution (0.2 M, pH 4.0), and the mixture was incubated at 35 °C for 10 min. Then, 50 uL AA solution (1.5 μM) were added into it and mixed uniformly. Their UV-visible absorption spectra were measured at 652 nm. After that, the mixture was re-incubated for 10 min and tested by an ultraviolet-visible spectrophotometer. The above process was repeated many times.

### 2.7. Detection of Ascorbic Acid (AA)

First, 40 uL TMB (5 mM) and 60 uL Pt NP solution (0.75 mg mL^−1^) were added into 2.8 mL acetic acid-sodium acetate buffer solution (0.2 M, pH 4.0). Then, the mixture was incubated at 35 °C for 10 min, followed by adding 50 uL AA solution. Their UV-visible absorption spectra were measured at 652 nm. Without centrifuging the catalyst, the solution after being tested was re-incubated at 35 °C, changes in ox-TMB concentration in the solution were monitored by the UV-visible spectrophotometer until the oxidation of the solution was comparable to the previous test, and the absorbance spectra was re-measured at 652 nm. This process was repeated multiple times.

## 3. Results and Discussion

### 3.1. Characterization of Pt Nanoparticles (Pt NPs)

As shown in Figure 1a, the prepared Pt NPs were about 30 nm and formed by these so-called “building blocks” with a size of 5 nm [28]. The size distributions of Pt NPs were analyzed based on counting more than 60 nanoparticles. As displayed in Figure 1b, the size of Pt NPs was relatively uniform in principle and concentrated around 28–32 nm. In addition, EDS analysis showed that the particles consisted of pure Pt (Figure 1c). From the analysis of high-resolution (HR)-TEM images, the lattice spacing of the Pt NPs was 0.196 nm and consistent with the (100) crystal plane of Pt nanoparticles, which was also consistent with literature reports and proved that nanoparticles with a stacked structure had been synthesized [28,29]. Pt nanoparticles with different morphologies were synthesized by changing the reaction conditions in order to evaluate the oxidase-like activities of the Pt NPs, as shown in Figure 2. By changing the HCl concentration during the preparation process, Pt-0 HCl of about 5 nm, Pt-3.7 HCl of about 9 nm, and Pt-9.25 HCl of about 16 nm formed by the accumulation of 7–8 nm Pt nanoparticles were prepared (Figure 2a–c). At the same time, the Pt-2 PVP was prepared by changing the PVP concentration, and it was found that Pt-2 PVP and Pt NPs were similar in morphology (Figure 2d). Their size distribution diagrams and EDS elemental analysis are shown in Appendix A, respectively.

### 3.2. Evaluation of the Oxidase-Like Catalytic Performances of Pt NPs

Using TMB as substrates, the oxidase-like activity of Pt nanoparticles was evaluated by testing the absorbance curve of the reaction system without adding H_2_O_2_. As shown in Figure 3a, Pt nanoparticles catalyzed the oxidation of TMB to ox-TMB, and the reaction system generated a typical blue product and had a typical absorption peak at 652 nm. However, no characteristic peaks after testing were observed in the system containing only TMB, indicating that Pt nanoparticles had good catalytic activity. In addition, no characteristic peak of Pt nanoparticle suspension appeared at 550–750 nm, demonstrating that it had no significant effect on subsequent reactions. At the same time, the catalytic activities of Pt nanoparticles with different morphologies were compared, as shown in Figure 3b. After 10 min of reaction, the catalytic activities of Pt-0 HCl, Pt-9.25 HCl, and Pt NPs were slightly better than those of Pt-3.7 HCl and Pt-2 PVP. At the same time, in the characterization at this stage, the performance difference between Pt-0 HCl and Pt NPs could not be distinguished. In general, all of them had brilliant activity.

### 3.3. Optimization of Different Experimental Conditions

Similar to other nanoenzymes, the properties of Pt nanomaterials are closely related to the reaction conditions [30]. The effects of pH, temperature, and reaction time on the reaction system were studied. The catalytic activity was low at 20 °C; as the temperature increased, the catalytic performance of the nanoparticles improved, reaching a maximum at 35 °C, and maintaining more than 70% of the highest activity in the range of 30–40 °C (Figure 4a), demonstrating the potential for use at room temperature. The pH also had a great impact on nanoenzyme performance, as shown in Figure 4b, where the performance was best at pH 4.0. The pH value above and below 4.0 both affected its oxidase-like activity, which is similar to natural enzymes [30,31]. Further research found that the absorbance of the ox-TMB solution increased with time, and the reaction rate gradually decreased after 8 min of reaction (Figure 4c). Therefore, in order to reduce the interference of environmental factors and the convenience of experiments, the following experiments used the following reaction conditions: Pt nanoparticle concentration 15 μg/mL, reaction temperature 35 °C, reaction pH 4.0, reaction time 10 min.

### 3.4. Steady-State Kinetic Assay of Pt NPs

In order to quantify the catalytic performance of the prepared Pt nanomaterials, the time course mode was used in this experiment to determine the reaction initial velocity and explore the kinetic parameters. By fixing other parameters and changing the TMB concentration, the absorbance change of the reaction solution at 652 nm three minutes before the reaction was tracked to calculate the concentration of the oxidized product by the Lambert−Beer law, and a kinetic curve was established. As shown in Figure 5a, the obtained curve satisfied the typical Michaelis–Menten equation, and the Lineweaver–Burk double inverse curve (inset Figure 5a) was obtained. The calculated maximum reaction rate was 330 nM S^−1^, and the Michaelis−Menten constant (Km) was 0.102 mM. Compared to the Km of many nanoenzymes that have been reported, the Km obtained was smaller (Appendix A), indicating that the affinity of PtNPs with TMB was great (a smaller Km indicates a higher affinity with the substrate [32,33]). Compared to the different nanoparticles prepared, as shown in Figure 5b, it was found that the Km of Pt NPs was slightly smaller than those of other nanoparticles. At the same time, the Km of Pt NPs was close to those of Pt-0 HCl and Pt-9.25 HCl, and the Km of Pt-3.7 HCl and Pt-2 PVP was slightly larger. This may be due to the large number of PVPs on the surface of Pt-2 PVP, which caused a large number of active sites to be occupied [34]. However, in general, the Km of these nanoparticles was not very different, which was consistent with previous tests.

### 3.5. Reusable Performance of Pt NPs

In order to further investigate the oxidase-like activity of Pt NPs, a cyclic experiment was designed (achieved by continuous redox TMB with AA and Pt NPs). As shown in Figure 6a, the reusability of Pt NPs for catalyzing the redox reactions was up to six cycles. We found that the performance of the second cycle redox reaction was similar to that of the first one, and the performance decreased about 10% at the third cycle redox reaction. After five cycles, its performance was greatly restored after the Pt NPs being centrifuged and washed with ethanol. The slight decrease in activity may be mainly caused by the loss of Pt NPs during the washing process. At the same time, Pt nanoparticles with different morphologies were compared to evaluate the reusability of Pt NPs. Figure 6b showed that except Pt NPs, the performance of other Pt nanoparticles was greatly reduced as the second cycle. In addition, Section 3.2 and Section 3.4 in this paper showed that the Km of various Pt nanoparticles prepared and the amount of ox-TMB produced after a full reaction in ten minutes were very close, but their reusability was quite different. Comparing Figure 1 and Figure 2a, Pt NPs were formed by 5 nm Pt nanoparticles, and Pt nanoparticles (Pt-0 HCl) were about 5 nm simply (Appendix A). Their catalytic performance was similar in the first cycle, and in the second cycle, their performance started to differ significantly. It is reported that the size of nanoparticles is an important factor to determine their enzyme-like activity [35,36]. Therefore, Pt-0 HCl exhibited a great oxidase-like activity in the first cycle. The Pt NPs, which were formed by the accumulation of 5 nm Pt nanoparticles, have similar activity to Pt-0 HCl during the first cycle, which may be due to the relatively reduced exposed active sites due to their stacked structure. In the second cycle, the difference in catalytic activities between Pt NPs and Pt-0 HCl became larger (Figure 6b). The main reason was probably that the stacked structure of Pt NPs was conducive to the substance transport and sufficient contact, so the performance was more stable [28,34]. Similarly, as shown in Figure 2b and Appendix A, the prepared Pt nanoparticles (Pt-3.7HCl) increased to 9 nm with poor dispersibility, and no “building blocks” existed, which had a slightly lower performance in the first cycle, and significantly lower performance in subsequent cycles, than Pt NPs and Pt-0 HCl (Figure 6b). In addition, compared to Pt-0 HCl, Pt-3.7 HCl had more defects, but its larger nanoparticles and worse dispersion led to degradation in the catalytic activity (Figure 2b, Appendix A, Figure 6b). Unlike Pt-0 HCl and Pt-3.7 HCl, so-called “building blocks” existed in Pt-9.25 HCl—approximately 16 nm nanoparticles were formed by small nanoparticles with the size of 7–8 nm, and not unexpectedly, the reusability performance of Pt-9.25 HCl was better than that of Pt-3.7 HCl (Figure 2c, Appendix A, Figure 6b). However, the reusability performance of Pt nanoparticles with the same morphology was not necessarily the same. As shown in Figure 2d and Appendix A, the Pt nanoparticles (Pt-2 PVP) had similar morphology and good dispersibility as Pt NPs. However, its reusability performance was lower than that of Pt NPs (Figure 6b). It may be due to the excessive PVP (as surface covering agent) added during the preparation process, so too many active sites were occupied (Appendix A showed that the surface of Pt NPs was covered with an amount of PVP). 

### 3.6. Detection of Ascorbic Acid (AA)

Without centrifuging the catalyst, the solution after being tested was re-incubated at 35 °C, and detection of the change in the concentration of ox-TMB in the reaction solution was carried out by an ultraviolet-visible spectrophotometer. After the solution was oxidized to a certain degree, the AA solution was added and tested immediately to determine the detection limit of the AA content of the colorimetric method. After three testing times, the linear calibration plots were obtained by analysis (Figure 7). It is found that the limits of detection (LOD) obtained were close, which were 131 ± 15, 144 ± 14, and 152 ± 9 nM, indicating that the detection accuracy had no significant change in at least three circles, which confirmed that Pt NPs had great reusability. In addition, the linear range of the three reactions was obtained. As shown in Figure 6, the linear range of the three reactions all falls approximately between 1 and 20 μM. At the same time, the selectivity of Pt NPs was tested, as shown in Appendix A, and it was found that the common ions had little effect on the test, indicating that the method have potential applications in general water quality testing and other fields.

## 4. Conclusions

In summary, a variety of Pt nanoparticles with different morphologies were prepared by changing the reaction conditions. Compared to Pt-0 HCl, Pt-3.7 HCl, Pt-9.25 HCl, Pt NPs, and Pt-2 PVP, it was found that the prepared Pt NPs had good intrinsic oxidase-like activity and reusability. The characteristic was mainly due to the so-called “building blocks” of Pt NPs—this structure allowed it to have a large specific surface area, and may be beneficial to material transfer [28]. In addition, by repeatedly detecting AA three times, we found that the limit of detection obtained did not change significantly. This study demonstrates the potential for reuse of Pt NPs, and provides an important direction for subsequent research.

## Figures and Tables

**Figure 1 nanomaterials-10-01015-f001:**
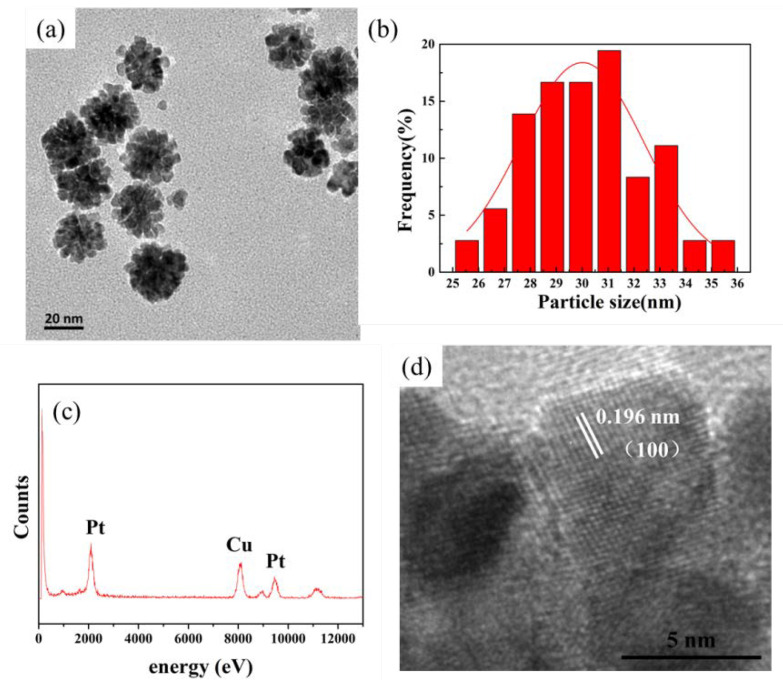
(**a**) TEM image of Pt nanoparticles (NPs), (**b**) size distributions of Pt NPs, (**c**) EDS elemental analysis of Pt NPs, and (**d**) HR-TEM image of Pt NPs.

**Figure 2 nanomaterials-10-01015-f002:**
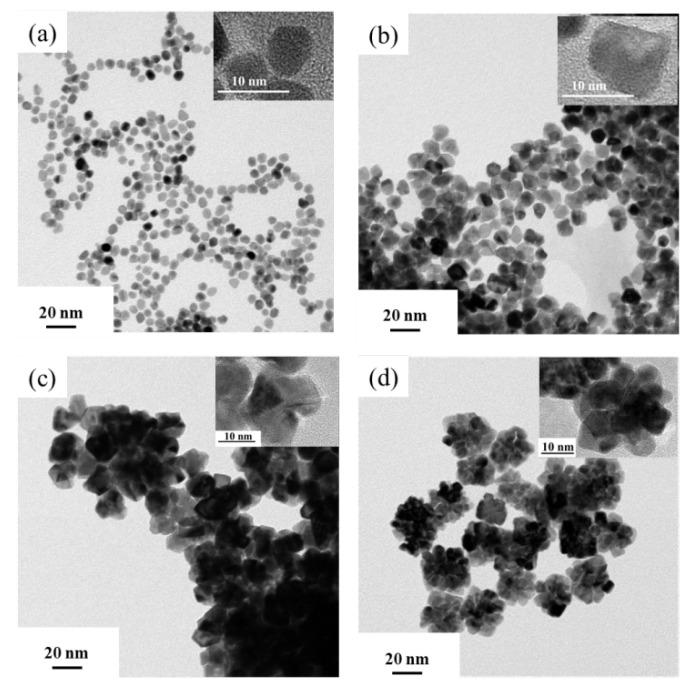
TEM images of Pt nanoparticles obtained by adding HCl and polyvinylpyrrolidone (PVP) with different concentrations: (**a**) Pt-0 HCl (without HCl), (**b**) Pt-3.7 HCl (3.7 wt% HCl), (**c**) Pt-9.25 HCl (9.25 wt% HCl), and (**d**) Pt-2 PVP (0.075 M PVP), Inset: TEM images of single Pt nanoparticle.

**Figure 3 nanomaterials-10-01015-f003:**
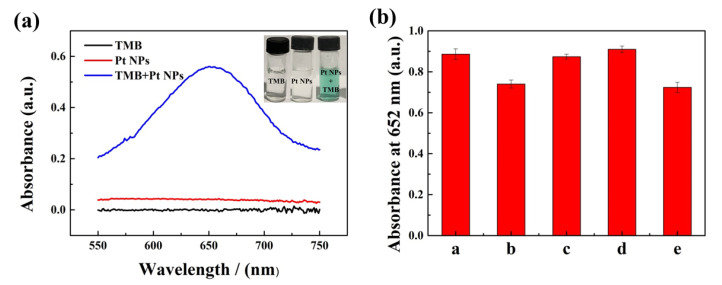
(**a**) UV-visible absorption spectra of different systems: 3, 3, 5, 5′-tetramethylbenzidine (TMB), Pt NPs, TMB + Pt NPs (reaction time: 5 min); (**b**) absorbance of the reaction systems at 652 nm with addition of different Pt nanoparticles (a: Pt-0 HCl, b: Pt-3.7 HCl, c: Pt-9.25 HCl, d: Pt NPs, e: Pt-2 PVP).

**Figure 4 nanomaterials-10-01015-f004:**
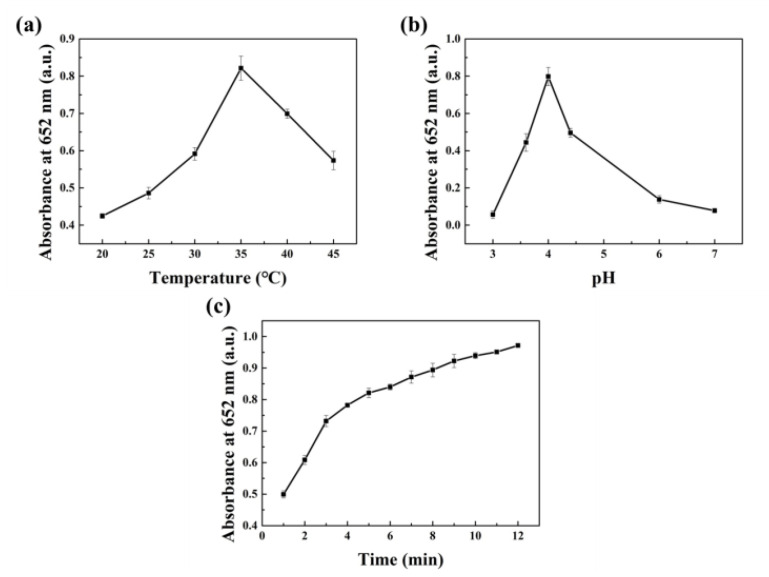
Effects of different conditions on the oxidase-like activity of Pt NPs: (**a**) Temperature, (**b**) pH, (**c**) time.

**Figure 5 nanomaterials-10-01015-f005:**
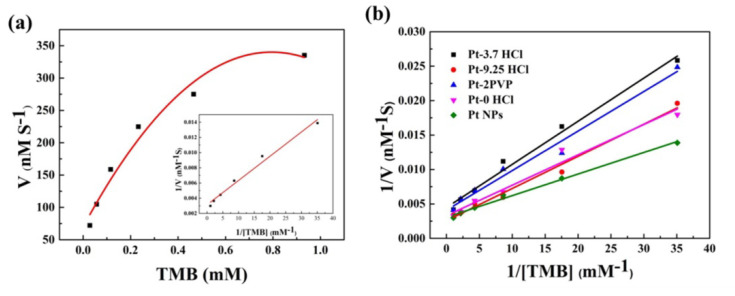
(**a**) Michaelis–Menten curve of Pt NPs with TMB, Inset: Lineweaver–Burk plots of Pt NPs with TMB; (**b**) Lineweaver–Burk plots of different Pt nanoparticles.

**Figure 6 nanomaterials-10-01015-f006:**
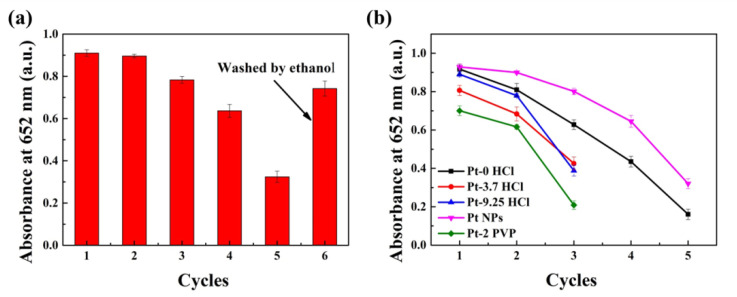
Reusability of (**a**) Pt NPs and (**b**) Pt nanoparticles with different morphologies for catalytically oxidizing TMB.

**Figure 7 nanomaterials-10-01015-f007:**
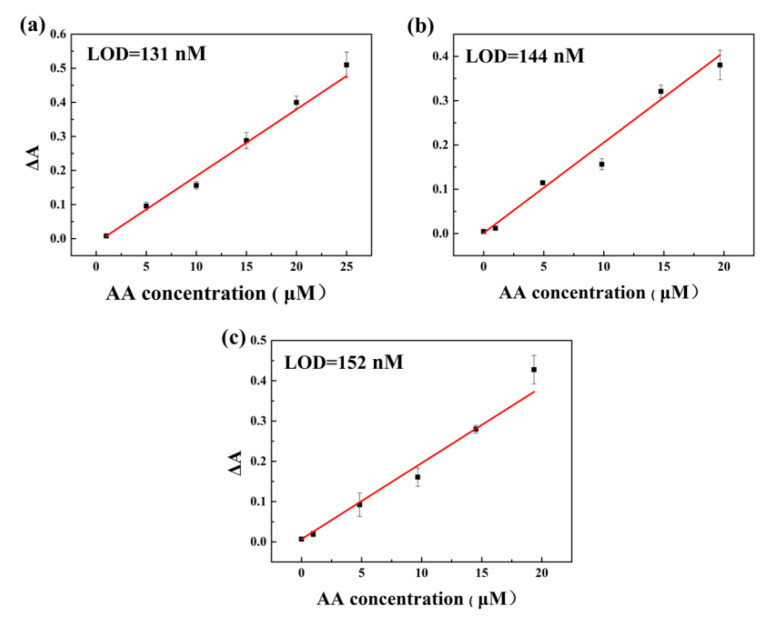
Linear calibration plot for three consecutive detections of ascorbic acid (AA): (**a**) the first time, (**b**) the second time, and (**c**) the third time.

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
