# Peer review of "Pt Nanoparticles with High Oxidase-Like Activity and Reusability for Detection of Ascorbic Acid"

_nanomaterials, 2020, doi:10.3390/nano10061015_

Round 1

Reviewer 1 Report

Yang et al. have reported a paper concerning the synthesis of Pt nanoparticles with different morphologies, obtained by changing the reaction conditions. These nanoparticles have been used as catalyst for the TMB oxidation, and the reaction was followed by colorimetric measurements at 652 nm. The possibility to detect ascorbic acid has been demonstrated, following the absorbance change at 652 nm.

The paper is quite well written, but at this stage is not publishable. I have some concerns, as follow:

  1. ABSTRACT: to my opinion, the abstract is not clear. It starts with a quotation:"Noble metal nanoenzyme has magnificent activity" (pag. 1, line 9) without describing what a metal nanoenzyme is. Also the quotation: "For the first time, we compared the Pt NPs which were formed by 5 nm Pt nanoparticles and Pt nanoparticles (Pt-0 HCl) which were about 5 nm simply." (pag. 1, lines 13-14) is very difficult to understand. What does it mean? Furthermore the acronym AA (pag. 1, line 19) in the abstract should be avoided.
  2. The oxidation reaction of TMB should be described, showing the specie responsible of the blue absorption at 652 nm.
  3. Do the authors have an idea describing the effect of the temperature and of the pH on the reaction?
  4. Figure 4a: I think there is an error: Figure 4 and Figure 5 are the same. Furthermore in Figure 4a there is no inset as reported in the paper. Figure 4 should be changed with the right one.
  5. The different NP morphology is an important parameter and should be added in the paper and not as supporting info.
  6. The error of the LOD should be reported, in order to compare the results.

Author Response

Response to Reviewer 1 Comments

Point 1: ABSTRACT: to my opinion, the abstract is not clear. It starts with a quotation:"Noble metal nanoenzyme has magnificent activity" (pag. 1, line 9) without describing what a metal nanoenzyme is. Also the quotation: "For the first time, we compared the Pt NPs which were formed by 5 nm Pt nanoparticles and Pt nanoparticles (Pt-0 HCl) which were about 5 nm simply." (pag. 1, lines 13-14) is very difficult to understand. What does it mean? Furthermore the acronym AA (pag. 1, line 19) in the abstract should be avoided.

Response 1: ABSTRACT has been modified:Noble metal nanoenzymes such as Pt,Au,Pd,etc. exhibit magnificent activity. However, due to the scarce reserves and expensive prices of precious metals, it is essential to investigate their enzyme –like activity and explore the possibility of their reuse. In this work, the oxidase-like activity and reusability of several Pt nanoparticles with different morphologies were detected. We compared the Pt NPs with the size of about 30 nm self-assembled by 5 nm Pt nanoparticles and Pt nanoparticles (Pt-0 HCl) with the diameter of about 5 nm, found that their Michaelis−Menten constant (Km) were close and initial performance was similar, but Pt NPs had better reusability. This probably attributed to the stacked structure of Pt NPs which was conducive to the substance transport and sufficient contact. At the same time, it’s found that the size, dispersion, and organic substances adsorbed on the surface of Pt nanoparticles would have a significant impact on their reusability. A colorimetric detection method was designed using the oxidase-like activity of Pt NPs to detect ascorbic acid in triplicate. The limits of detection were 131±15 nM, 144±14 nM and 152±9 nM, with little difference. This research not only showed that the morphology of the catalyst could be changed and its catalytic performance could be controlled by a simple liquid phase synthesis method, but also had great significance for the reuse of Pt nanoenzymes in the field of bioanalysis.

Point 2: The oxidation reaction of TMB should be described, showing the specie responsible of the blue absorption at 652 nm.

Response 2: The oxidation reaction of TMB is as follows[1]:

Literature references of this oxidation reaction and products have been added to the paper (pag5.line158).

Point 3: Do the authors have an idea describing the effect of the temperature and of the pH on the reaction?

Response 3: Several reports have studied the principle of nanozyme-like activity and the effect of temperature or pH on the catalytic activity of noble metal nanozymes (Au, Pt, Pd, etc.)

  Reportedly, single-atomic O adatoms on metal surfaces have Brønsted-base character. They are able to abstract acidic hydrogens from surrounding molecules to act as oxidizing agents. Therefore, the following two reactions (eq 1 and eq 2) serve as a plausible mechanism for the oxidase-like activities of metals. This process is greatly affected by environmental factors such as pH and temperature[2,3]. But the specific principle remains to be studied.

O2=2O*   (1)

O*+S=H2O*+Sox   (2)

Point 4: Figure 4a: I think there is an error: Figure 4 and Figure 5 are the same. Furthermore in Figure 4a there is no inset as reported in the paper. Figure 4 should be changed with the right one.

Response 4: I'm sorry that I made a mistake about Figure 4 and 5. The original Figure 4 has been changed with the right one, now it is Figure 5 (pag7. line190-192). The original Figure 5 is correct (pag7. Line 210-212).

Point 5: The different NP morphology is an important parameter and should be added in the paper and not as supporting info.

Response 5: The morphology of different Pt nanoparticles has been moved to the paper, as shown in Figure 2 (pag5. Line 150-153).

Point 6: The error of the LOD should be reported, in order to compare the results.

Response 6:  It has been optimized :LOD1=131±15 nM;LOD2=144±14 nM; LOD3=152±9 nM. (pag9. Line 260)

References.

[1] Qiang, Hong, Li, et al. A sensitive Hg(II) colorimetric sensor based on synergistic catalytic effect of gold nanoparticles and Hg[J]. Sensors & Actuators B Chemical, 2016.

[2] Xiaomei, Shen, Wenqi, et al. Mechanisms of Oxidase and Superoxide Dismutation-like Activities of Gold, Silver, Platinum, and Palladium, and Their Alloys: A General Way to the Activation of Molecular Oxygen[J]. Journal of the American Chemical Society, 2015.

[3] Lin Y , Ren J , Qu X . Nano-Gold as Artificial Enzymes: Hidden Talents[J]. Advanced Materials, 2014, 26(25):4200-4217.

Reviewer 2 Report

This is paper in which Pt nanoparticles are deployed for ascorbic acid detection. The preparation of the particles and optimisation of the detection conditions using the oxidation of tetramethylbenzidine as a chromogenic substrate are described. The use of Pt nanoparticles for this type of detection is certainly topical but this paper is poorly put together and difficult to understand in parts what the authors are trying to say. There are a number of deficiencies that the authors have to address to bring the paper to a satisfactory standard:

Main comments  

The abstract has to re-write to make clear the procedure and the main findings of the investigations.

The whole paper would require extensive editing to improve clarity

The introduction has to be re-written so that the main focus is on the use of Pt nanoparticles as nanoenzymes.

The description of the process for the production of the Pt nanoparticles is unclear, and should be re-written.

Provide the evidence to clearly show the formation of the different PtNPs morphologies. It is difficult to identify the differences in the morphologies of the particles. The authors should point at the distinguishing features.

Minor comments

Line 9 should read: Noble metal nanoenzymes have excellent activity

Line 10 should read: ….. it is essential to investigate their enzyme –like activity and explore the possibility of their reuse.

Line 44 should read: … investigations into their mechanisms of action are topical.

Line 48 should read:  Pt is an expensive precious metal.

Line 49 should read: It is essential that the PtNPs are recycled after use In order to conserve supplies.

Line 53 should read: These investigations extend our understanding…..

Line 57 Write the formula as: H2PtCl6.6H2O.

Line 116 should read: Evaluation of the oxidase-like activities of the PtNPs

Line 128-130: Make clear the difference between Pt=0HCl and PtNPs

Line 138-139: The phrase : maintained at a high level does not make sense when one examines the plot. Re-write to explain what is observed.

Line 142-143: The absorbance of Ox-TMS shows continuous increase with time with no levelling off. Should include plots from longer runs if they are available.

Line 149: Cannot see the inset.  Where is the Lineweaver-Burk plot?

Line 153-156: Re-write to make clear what you are trying to say here.

Line 169: I cannot see any difference between Figs 4 and 5.

Line 203: It would have been useful to provide evidence for the surface covering you claim to occur.

Line 228: Give some indication of why the chosen ions were tested.

Author Response

Response to Reviewer 2 Comments

Main comments

Point 1: The abstract has to re-write to make clear the procedure and the main findings of the investigations.

Response 1: Noble metal nanoenzymes such as Pt,Au,Pd,etc. exhibit magnificent activity. However, due to the scarce reserves and expensive prices of precious metals, it is essential to investigate their enzyme –like activity and explore the possibility of their reuse. In this work, the oxidase-like activity and reusability of several Pt nanoparticles with different morphologies were detected. We compared the Pt NPs with the size of about 30 nm self-assembled by 5 nm Pt nanoparticles and Pt nanoparticles (Pt-0 HCl) with the diameter of about 5 nm, found that their Michaelis−Menten constant (Km) were close and initial performance was similar, but Pt NPs had better reusability. This probably attributed to the stacked structure of Pt NPs which was conducive to the substance transport and sufficient contact. At the same time, it’s found that the size, dispersion, and organic substances adsorbed on the surface of Pt nanoparticles would have a significant impact on their reusability. A colorimetric detection method was designed using the oxidase-like activity of Pt NPs to detect ascorbic acid in triplicate. The limits of detection were 131±15 nM, 144±14 nM and 152±9 nM, with little difference. This research not only showed that the morphology of the catalyst could be changed and its catalytic performance could be controlled by a simple liquid phase synthesis method, but also had great significance for the reuse of Pt nanoenzymes in the field of bioanalysis.

Point 2: The whole paper would require extensive editing to improve clarity.

Response 2: The whole paper has been edited: In order to express more clearly, the morphology of different Pt nanoparticles in supporting info has been moved into the paper. And added morphology and infrared analysis.(pag5.line 150-153) (pag8.line 249)

Point 3: The introduction has to be re-written so that the main focus is on the use of Pt nanoparticles as nanoenzymes.

Response 3: The introduction has been optimized: Added description of various enzyme activities of Pt nanomaterials. (pag2. line 61-78)

Point 4: The description of the process for the production of the Pt nanoparticles is unclear, and should be re-written.

Response 4: The description of the process for the production of the Pt nanoparticles has been re-written: In a 50 mL three-necked flask, 2.5 mL of ethylene glycol (EG) was refluxed at 180 oC for 5 minutes with oil bath. After which, 0.34 mL of HCl (37 wt%) was added. Then, 0.94 mL of H2PtCl6·6H2O (0.0625 M) solution and 4 mL of PVP (0.0375 M) were added into the above solution for 10 times within 5 minutes. During the all reaction, the reactants were continuously stirred and mixed uniformly. After 20 minutes, the three-necked flask was taken out, naturally cooled to room temperature. The sample was collected and washed by centrifugation with water and ethanol. Finally, the samples was dried in an oven at 60 oC, got Pt NPs. By changing the concentration of HCl (37 wt%) into 0 wt%, 3.7 wt% and 9.25 wt%,, prepared Pt-0 HCl (0 wt%), Pt-3.7 HCl (3.7 wt%), Pt-9.25 HCl (9.25 wt%),respectively. And changing the concentration of PVP solution (0.00375 M) into 0.075 M, prepared Pt-2PVP (0.075 M). (pag3. line 98-102)

Point 5: Provide the evidence to clearly show the formation of the different Pt NPs morphologies. It is difficult to identify the differences in the morphologies of the particles. The authors should point at the distinguishing features.

Response 5: Added Figure 2 and related descriptions in the paper to characterize different Pt nanoparticles. (pag4.line 143-149) (pag5. line 150-153)

Minor comments

Point 1: Line 9 should read: Noble metal nanoenzymes have excellent activity.

Response 1:It has been revised. (pag1. line 23)

Point 2: Line 10 should read: ….. it is essential to investigate their enzyme –like activity and explore the possibility of their reuse. 

Response 2:It has been revised. (pag1. line 25)

Point 3: Line 44 should read: … investigations into their mechanisms of action are topical. 

Response 3:It has been revised. (pag2. line 59)

Point 4: Line 48 should read:  Pt is an expensive precious metal. 

Response 4:It has been revised. (pag2. line 65)

Point 5: Line 49 should read: It is essential that the Pt NPs are recycled after use In order to conserve supplies. 

Response 5:It has been revised. (pag2. line 66-67)

Point 6: Line 53 should read: These investigations extend our understanding…

Response 6:It has been revised. (pag2. line 76-77)

Point 7: Line 57 Write the formula as: H2PtCl6.6H2O.

Response 7:It has been revised. (pag2. line 81)

Point 8: Line 116 should read: Evaluation of the oxidase-like activities of the PtNPs

Response 8:It has been revised. (pag4. line 143-144)

Point 9: Line 128-130: Make clear the difference between Pt=0 HCl and Pt NPs

Response 9:It has been revised: In the characterization at this stage, the performance difference between Pt-0 HCl and Pt NPs could not be distinguished. (pag5.line 165-166)

(The performance difference between Pt-0 HCl and Pt NPs was mainly reflected in their different reusability).

Point 10: Line 138-139: The phrase: maintained at a high level does not make sense when one examines the plot. Re-write to explain what is observed.

Response 10:It has been revised: As the temperature increasing, the catalytic performance of the nanoparticles improved, reached a maximum at 35 °C and maintained more than 70% of the highest activity in the range of 30-40 °C. (pag6.line 178-179)

Point 11: Line 142-143: The absorbance of Ox-TMB shows continuous increase with time with no levelling off. Should include plots from longer runs if they are available.

Response 11:In order to determine the best reaction conditions and facilitate the experiment, the best reaction time was tested. The reaction rate gradually decreased after 8 minutes of reaction was observed, and the reaction time was continued to be extended. The small range of absorbance continued to increase and had little effect on the overall experiment. And I'm sorry I don't have more data.

Point 12: Line 149: Cannot see the inset.  Where is the Lineweaver-Burk plot?

Response 12:I'm sorry that I made a mistake about Figure 4 and 5. The original Figure 4 has been changed with the right one, now it is Figure 5 (pag7.line 190-192). The original Figure 5 is correct.

Point 13: Line 153-156: Re-write to make clear what you are trying to say here.

Response 13:I made a mistake about Fig4, it has been revised. Fig 5 now shows the Michaelis–Menten curve and Lineweaver-Burk plots accurately. Now it should have made it clear (pag7.line190-192).

Point 14: Line 169: I cannot see any difference between Figs 4 and 5.

Response 14:It has been revised. (pag7.line190-192).

Point 15: Line 203: It would have been useful to provide evidence for the surface covering you claim to occur.

Response 15:Infrared analysis has been added and as shown in Figure S3.

Point 16: Line 228: Give some indication of why the chosen ions were tested.

Response 16:There is no such item in line 228. Line 223 is suspected and has been revised: It was found that the common ions had little effect on the test,indicating that the method had potential applications in general water quality testing and other fields. (pag9. line 265-266).

Round 2

Reviewer 2 Report

Points 5 and 16 are yet to be adequately addressed. I can not see the differences between the Pt particles. Can the authors point to the differences between the Pt particles. The images from the electron microscope all look similar. How can one distinguish one from the other?

Why did you select Na, K, Ca and the other ions for use in the interference study? 

Author Response

Response to Reviewer 2 Comments

Point 1: I can not see the differences between the Pt particles. Can the authors point to the differences between the Pt particles. The images from the electron microscope all look similar. How can one distinguish one from the other? 

Response 1: In order to make the difference in morphology of different nanoparticles more clear, I modified Figure 2, added TEM images of single Pt nanoparticle. As shown in Figure 2, Pt-0 HCl were about 5 nm,Pt-3.7 HCl were about 9 nm ,and Pt-9.25 HCl were about 16 nm formed by the accumulation of 7-8 nm Pt nanoparticles with poor dispersion, full of defects. And it was found that Pt-2 PVP and Pt NPs were similar in morphology, both were about 30 nm and formed by these so-called “building blocks” with size of 5 nm [1](Figure 2, Figure 1). (pag4-5. line 144-154)

Point 2: Why did you select Na, K, Ca and the other ions for use in the interference study?

Response 2: Common ions such as K +, Na +, etc. are widely present in water, beverages, etc. Exclude their influence on the test, can show that the test method has certain feasibility. And of course, if we want to explain the universality of the method, the more ions and substances are tested, the better

[1]Cao, Y.; Yang, Y.; Shan, Y.; Fu, C.; Nguyen Vet, L.; Huang, Z.; Guo, X.; Nogami, M. Large-scale template-free synthesis of ordered mesoporous platinum nanocubes and their electrocatalytic properties. Nanoscale 2015, 7, 19461-19467. doi:10.1039/c5nr05772h.
